# Joint Homophily and Heterophily Relational Knowledge Distillation for Efficient and Compact 3D Object Detection

## ABSTRACT

3D Object Detection (3DOD) aims to accurately locate and identify 3D objects in point clouds, facing the challenge of balancing model performance with computational efficiency. Knowledge distillation emerges as a vital method for model compression in 3DOD, transferring knowledge from complex, larger models to smaller, efficient ones. However, the effectiveness of these methods is constrained by the intrinsic sparsity and structural complexity of point clouds. In this paper, we propose a novel methodology termed Joint Homophily and Heterophily Relational Knowledge Distillation (H2RKD) to distill robust relational knowledge in point clouds, thereby enhancing intra-object similarity and refining inter-object distinction. This unified strategy encompasses the integration of Collaborative Global Distillation (CGD) for distilling global relational knowledge across both distance and angular dimensions, and Separate Local Distillation (SLD) for a focused distillation of local relational dynamics. By seamlessly leveraging the relational dynamics within point clouds, the H2RKD facilitates a comprehensive knowledge transfer, significantly advancing 3D object detection capabilities. Extensive experiments on KITTI and unScenes datasets demonstrate the effectiveness of the proposed H2RKD.

## CCS CONCEPTS

• **Computing methodologies → Object detection**; **Computer vision representations**.

## KEYWORDS

3D Object Detection, Relational Knowledge Distillation

**ACM Reference Format:**

Anonymous Author(s). 2024. Joint Homophily and Heterophily Relational Knowledge Distillation for Efficient and Compact 3D Object Detection. In *Proceedings of the 32nd ACM International Conference on Multimedia (MM'24), October 28-November 1, 2024, Melbourne, Australia.* ACM, New York, NY, USA, 9 pages. https://doi.org/10.1145/nnnnnnn.nnnnnnn

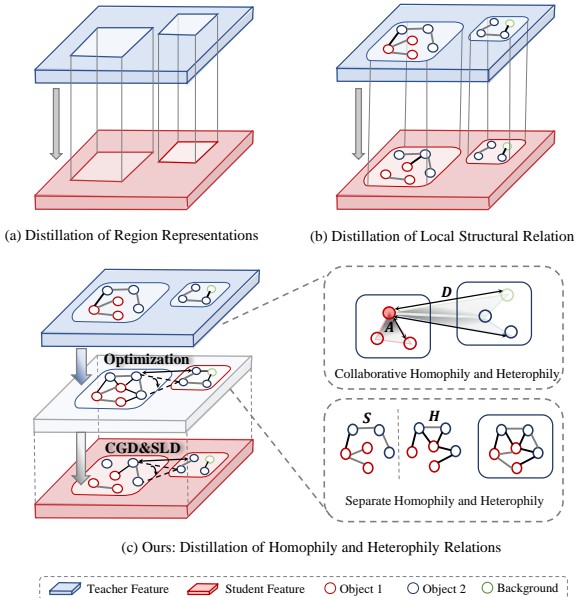

(a) Distillation of Region Representations

(b) Distillation of Local Structural Relation

(c) Ours: Distillation of Homophily and Heterophily Relations

**Figure 1: Comparison of different distillation methods. (b) acquires structural knowledge of the point clouds compared to (a). Our (c) refines the incomplete relationships obtained in (b), incorporating both homophily and heterophily relations.**

## 1 INTRODUCTION

Lidar-based 3D Object Detection (3DOD) [20, 30], leveraging point clouds for precise object identification and location, is a fundamental task in computer vision. Offering depth information beyond 2D detection, it's crucial for applications in robotics, autonomous vehicles, and augmented reality. However, increased detection capabilities come with higher computational costs. To mitigate this, Knowledge Distillation (KD) has been applied to balance performance with computational efficiency, transferring knowledge from larger to smaller models.

Despite advancements in Knowledge Distillation (KD)[6, 28, 36] for streamlining image understanding models, their application to 3D Object Detection (3DOD) faces significant challenges. The primary obstacles stem from the inherent characteristics of point clouds, such as sparsity, irregularity, and geometric complexity, which hinder the generalization capabilities of these KD methods. To alleviate these issues, previous work [13] has focused on reducing the representational gap between teacher and student models by enhancing mutual information across pairs of regions, as illustrated in Figure 1(a). To boost the transfer of structural information of objects, the other works tend to further distill relational knowledge. Concisely, PointDistiller [32] aims to distill the local geometric structures captured in K-Nearest Neighbor(KNN) graphs to uphold neighborhood relations, as shown in Figure 1(b). However, the graph construction, heavily based on homophily, concentrates the learning within similar classes failing to embrace the multiple structural relationships present. More specifically, as Figure 2 demonstrates, the features extracted by the teacher show obvious

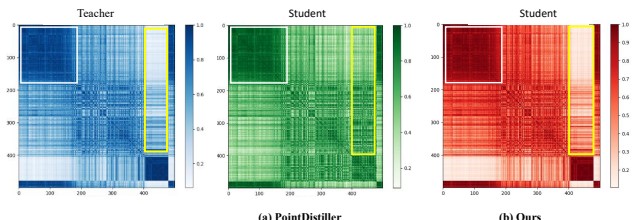

**Figure 2: Visualization of the relations among voxel features from the teacher and the student distilled by (a) and (b) on KITTI, respectively. The relations are demonstrated by the similarity matrix, where the white box represents high similarity, and the yellow box represents weak similarity.**

similarity and difference relations, as marked by white and yellow boxes respectively. Unfortunately, after distillation, PointDistiller [30] only retains high similarity relations and significantly destroys most difference relations between voxel features. In contrast, our method emphasizes simultaneously distilling both relations to effectively imitate the teacher.

Obviously, a robust relational knowledge should encompasses both *homophily* and *heterophily*, *i.e.*, similarity relations within the same object and difference relations between different objects, respectively beneficial for enhancing the intra-class consistency and inter-class discrimination. Driven by the above analysis, in this paper, we propose a novel Joint Homophily and Heterophily Relational Knowledge Distillation method (H2RKD) for lidar-based 3D object detection, as illustrated in Figure 1(c). H2RKD models and transfers relational knowledge in two ways, *i.e.*, collaborative global distillation module (CGD) which distills global relations simultaneously, and separate local distillation module (SLD) which distills local homophily and heterophily separately. Specifically, CGD models distance-wise relations between pairs and angle-wise relations among triplets of features, implicitly collaborating both homophily and heterophily into global relations. Then two kinds of global relation consistency losses, including distance-wise relational knowledge distillation loss and angle-wise relational knowledge distillation loss, distill long-range semantic correlations buried in point clouds. Furthermore, to capture subtle dynamic local relations, SLD is proposed to separately embed local structure information into homophilic graphs and heterophilic graphs. Subsequently, SLD encodes and propagates intra-class and inter-class relations in dynamic graphs. Finally, a local relational knowledge distillation loss is adopted to distill local semantic relations and geometry information from teacher to student. Through the collaboration of CGD and SLD, our student model comprehensively learns the relational knowledge from the teacher, which preserves structural relations of point clouds, enhancing intra-class similarity and promoting inter-class discrimination simultaneously. We conduct extensive experiments on KITTI and large-scale nuScenes datasets to verify the effectiveness of the proposed approach.

The main contributions can be summarized as follows:

- We propose a novel Joint Homophily and Heterophily Relational Knowledge Distillation method (H2RKD) to distill

robust relational knowledge in point clouds, thereby enhancing intra-object similarity and refining inter-object distinction.
- We explore transferring homophily and heterophily relational knowledge in two ways: Collaborative Global Distillation module (CGD) distills global relational knowledge across both distance and angular dimensions, and Separate Local Distillation module (SLD) distills subtle local correlations and differences.
- Extensive experiments demonstrate the effectiveness of the above contributions, and our proposed method achieves state-of-the-art performance on the challenging 3DOD task.

## 2 RELATED WORK

### 2.1 Lidar-based 3D Object Detection

Lidar-based 3D object detection aims to localize and classify 3D objects from point clouds. These methods can be briefly categorized as point-based [3, 21, 29], pillar-based[12, 25], and voxel-based[7, 38]. (1)Point-based methods[21] leveraged pointnet [17] or pointnet++ [18] to extract sparse point features for 3D object detectors. (2)As a Pillar-based method, [12] utilized the pointnet to learn the representations of multiple-pillar point clouds, and convert these representations into a pseudo image, which can be processed with 2D convolutional layers. (3)For the voxel-based method, [38] proposed a single-stage detector that divides point clouds into equally spaced 3D voxels and processes them with voxel feature encoding layers.

Recently, some methods combining the aforementioned approaches have achieved notable performance, such as [20, 30]. However, as performance improves, lidar-based 3D detection models are likely to bury heavier computation costs. Hence, in this work, we focus on exploring knowledge distillation methods to boost the performance of lightweight 3D detectors.

### 2.2 Knowledge Distillation for Lidar-based 3D Object Detection

Knowledge distillation was originally proposed for model compression in [1] and focused on emulating the knowledge derived from a teacher network. Recently, knowledge distillation methods have demonstrated significant advancements in 2D object detection [23, 24, 33] and have also been leveraged to transfer knowledge in multi-modality setup [11, 37] or multi-frame to single-frame setup [35] in the 3D detection area.

Existing knowledge distillation methods in lidar-based 3D detection tend to transfer knowledge representations or structural relationships within point clouds. In the first line, most methods prioritize transferring knowledge in crucial regions to acquire robust knowledge representations. [27] leverages cues in teacher prediction to determine the important areas for distillation. [13] maximizes the mutual information between intermediate features by bilaterally transferring. Another line is embedded with transferring structural relationships to learn more discriminative representations. [4] distill 3D representation under the consideration of the correlation among the multiple detection head components.[32] encodes the semantic information in the local geometric structure based on a local topology map using KNN.

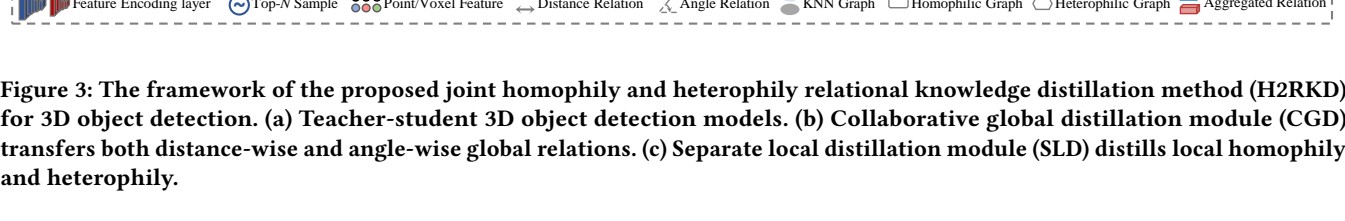

**Figure 3: The framework of the proposed joint homophily and heterophily relational knowledge distillation method (H2RKD) for 3D object detection. (a) Teacher-student 3D object detection models. (b) Collaborative global distillation module (CGD) transfers both distance-wise and angle-wise global relations. (c) Separate local distillation module (SLD) distills local homophily and heterophily.**

Nevertheless, current methods often focus solely on the relationships among objects or the structural relationship within locally similar point clouds, overlooking the global relationships. Therefore, we delve into global relationships and seek more comprehensive local structural relationships.

## 3 METHOD

### 3.1 Formulation and Overview

**Problem Formulation.** Given a set of point clouds $X = \{x_1, x_2, \ldots, x_n\}$ and the corresponding ground truth labels (GT labels) $Y = \{y_1, y_2, \ldots, y_m\}$, the 3D object detector can be formulated as $D = L \circ H$, including feature encoding layers $L$ and detection head $H$ for prediction. In this paper, our goal is to train a student detector $S$ under the supervision of a pre-trained teacher detector $T$, which is optimized by both 3DOD task loss and KD loss.

**Framework Overview.** Figure 3 illustrates the architecture of the proposed H2RKD method, comprising three key components: teacher-student 3D object detection models, collaborative global distillation (CGD) module, and separate local distillation (SLD) module. During training, teacher and student models extract two sets of point cloud features, $\tilde{F}_T = \{t_i\}_{i=1}^{M}$ and $\tilde{F}_S = \{s_i\}_{i=1}^{M}$, for total $M$ points/voxels. Based on $\tilde{F}_S$, the student detection head predicts detection results, for calculating task loss and evaluation. To enhance feature robustness and simplify the relation complexity in $\tilde{F}_T$ and $\tilde{F}_S$, we follow [32] to sample $N$ features $F_T = \{t_i\}_{i=1}^{N}$ and $F_S = \{s_i\}_{i=1}^{N}$. Subsequently, relational knowledge is distilled in two ways: (1) CGD represents distance-wise and angle-wise relations within $F_T$ or $F_S$ by the Global Relation Modeling (GRM) module and

distills global relations through distance-wise RKD loss and angle-wise RKD loss, respectively. (2) SLD first constructs KNN graphs to model local relations within $F^T$ and $F^S$, reconstructs them into homophilic graphs and heterophilic graphs by Separate Relation Modeling (SRM), and encodes and propagates relations within each graph. Then SLD aggregates homophily and heterophily within $F_T$ or $F_S$ and employs local RKD loss to distill these local relations. Finally, the overall framework is jointly optimized with 3DOD task loss, two global RKD losses, and a local RKD loss. During inference, the trained student model predicts detection results for performance evaluation.

Subsequently, we provide a detailed description of the CGD module, SLD module and loss functions.

### 3.2 Collaborative Global Distillation

Collaborative global distillation implicitly embeds homophily and heterophily into two global relations, including second-order and third-order relations within point clouds. Specially, the second-order relation is modeled by the distance-wise relation between pairs of features and the third-order relation is modeled by the angle-wise relation among ternary features.

Drawing inspiration from the concept proposed in [15], we employ two straightforward yet effective potential functions to capture these global relations, considering distance-wise and angle-wise relations. Correspondingly, we propose the distance-wise RKD loss and angle-wise RKD loss to distill the global relations.

**Global Relation Modeling.** We define $\psi$ as a relational potential function for modeling each $\chi^N$, where $\chi^N = \{(x_i, \ldots, x_j) | i, j \in N, i \neq j\}$ represents a set of $N$-tuples of distinct data. Thus, we

denote the second-order relation in $F_T$ as $\chi_T^2 = \{(t_i, t_j)|i \neq j\}$ and in $F_S$ as $\chi_S^2 = \{(s_i, s_j)|i \neq j\}$, respectively. Similarly, we denote the third-order relation in $F_T$ as $\chi_T^3 = \{(t_i, t_j, t_k)|i \neq j \neq k\}$ and in $F_S$ as $\chi_S^3 = \{(s_i, s_j, s_k)|i \neq j \neq k\}$. Then we model pairwise and ternary relations of $F$ by distance-wise and angle-wise relations.

(1) Given a pair of point/voxel features $\chi_T^2$ from $F_T$, distance-wise potential function $\psi_D$ measures the Euclidean distance between the two features in the representation space:

$$\psi_D(t_i, t_j) = \frac{1}{\mu}\|t_i - t_j\|_2,$$
$$\mu = \frac{1}{|F_T|} \sum_{(t_i,t_j)\in\chi_T^2} \|t_i - t_j\|_2. \tag{1}$$

where $\mu$ is a normalization factor for distance. To focus on relative distances among other pairs, we set $\mu$ to be the average distance between pairs from $\chi^2$ in the mini-batch.

(2) Given a triplet of point/voxel features $\chi_T^3 = \{(t_i, t_j, t_k)|i \neq j \neq k\}$ from $F_T$, an angle-wise relational potential measures the angle formed by the three examples in the representation space:

$$\psi_A(t_i, t_j, t_k) = \langle e^{ij}, e^{kj}\rangle, \tag{2}$$
$$\text{where } e^{ij} = \frac{t_i-t_j}{\|t_i-t_j\|_2}, e^{kj} = \frac{t_k-t_j}{\|t_k-t_j\|_2}.$$

**Global Relation Distillation Loss.** We leverage two kinds of global relation consistency losses: Distance-wise RKD loss and Angle-wise RKD loss to transfer global relations for improving the perception of similarity and distinction in point clouds.

(1) Distance-wise relation is measured in both the teacher and the student, a distance-wise distillation loss is defined as:

$$\mathcal{L}_D = \sum l_\delta \left( \psi_D(t_i, t_j), \psi_D(s_i, s_j) \right), \tag{3}$$
$$\text{where } (t_i, t_j) \in \chi_T^2 \text{ and } (s_i, s_j) \in \chi_S^2.$$

(2) Angle-wise relation is measured in both the teacher and the student, an angle-wise distillation loss is defined as:

$$\mathcal{L}_A = \sum l_\delta \left( \psi_D(t_i, t_j, t_k), \psi_D(s_i, s_j, s_k) \right). \tag{4}$$

where $(t_i, t_j, s_j) \in \chi_T^3$ and $(s_i, s_j, s_k) \in \chi_S^3$. $l_\delta$ is Huber loss [10].

The distance-wise distillation transfers the relationship of examples by penalizing distance differences between their representation, while the angle-wise distillation loss transfers the relationship of training example embeddings by penalizing angular differences. Therefore, the CGD implicitly distills both homophily and heterophily into global relations.

## 3.3 Separate Local Distillation

Separate Local Distillation explicitly model local homophily and heterophily relations by homophilic graphs and heterophilic graphs. SLD encodes and propagates relations in each graph by a mixed filter and dynamic graph convolutional layers, respectively. Correspondingly, we propose the local knowledge distillation loss to distill the local relations.

**Separate Relation Modeling.** Define graph data as $G = \{\mathcal{V}, \tilde{A}, f\}$, where $\mathcal{V}$ represents the set of $n$ nodes, $f$ is the feature matrix from $F_T$ or $F_S$. We initialize graph structure $\tilde{A}$ based on these voxels or points clustered by KNN (K-Nearest Neighbours). The normalized adjacency matrix is $A = D^{-\frac{1}{2}}(\tilde{A} + I)D^{\frac{1}{2}}$, where $D$ represents the

degree matrix. The corresponding graph Laplacian is $L = I - A$. $\mathbf{1}.$is a matrix with all 1. Subsequently, we will provide a detailed description of constructing two graphs.

(1) Firstly, we construct a heterophilic graph by selecting the nodes that are far away from each other in both feature space and structure space as negative pairs. Specifically, we use complementary graphs of similarity graph $\overline{W}$ and topology graph $\overline{A}$ to construct a heterophilic graph $\mathcal{H}$. The procedure is formulated as follows:

$$\overline{W} = \mathbf{1}. - W,$$
$$\overline{A} = \mathbf{1}. - A, \tag{5}$$
$$\mathcal{H} = \overline{W} \odot \overline{A}.$$

where the similarity matrix $W$ is obtained through the cosine similarity of node features, which characterizes the closeness among nodes in feature space $f$. $\odot$ represents the Hadamard product, which is used to describe non-neighbor relations in both feature space and topology space.

(2) Simultaneously, we could further improve the homophily level of the raw graph by minimizing the distances among adjacent nodes, which is formulated as:

$$\min_{\mathcal{S}_{i:}} \sum_{j=1}^N \mathcal{S}_{ij}\|t_i - t_j\|_2 + \mathcal{S}_{ij}^2, \tag{6}$$

where $\mathcal{S}_{i:}$represents the i-th row of $\mathcal{S}$. Graph $\mathcal{S}$ will be more homophilic when edges are defined by nodes sharing high similarity. Furthermore, we use a regularization term to integrate the 1-hop and 2-hop neighbor relations. Let $\|t_i - t_j\|_2 = K_{ij}$, then we construct a homophilic graph $\mathcal{S}$ by solving the following optimization problem:

$$\min_{\mathcal{S}_{ij}} \mathcal{S}_{ij}K_{ij} + \mathcal{S}_{ij}^2 + (\mathcal{S}_{ij}^{(2)} - \mathcal{S}_{ij})^2,$$
$$s.t. \quad \mathcal{S}_{ij} > 0, \sum_{j=1}^N \mathcal{S}_{ij} = 1 \tag{7}$$

where $\mathcal{S}^{(2)}$ is the 2-hop graph, i.e.,$\mathcal{S}^{(2)} = \mathcal{S} \times \mathcal{S}$.

**Extracting Dynamic Local Relation.** We introduce a graph data mixture filter designed to handle diverse types of graphs, as follows:

$$\mathcal{F} = \beta(\frac{1}{2}L_\mathcal{H})^k f + (1-\beta)(I - \frac{1}{2}L_\mathcal{S})^k f. \tag{8}$$

where $L_\mathcal{S}$ and $L_\mathcal{H}$ are the normalized Laplacian matrices of reconstructed homophilic and heterophilic graphs, and $f$ is the feature matrix. Then, we apply a dynamic graph convolution $\mathcal{G}$ as the aggregation operation upon the final representation $\mathcal{F}$ to align the dimensions of the $\mathcal{F}_T$ and $\mathcal{F}_s$.

**Local Relation Distillation Loss.** With the reweighting strategy, the local relation distillation can be formulated as:

$$\mathcal{L}_{local} = \frac{1}{n} \sum_{i=1}^n \phi_i \cdot \|\mathcal{G}_i^T(\mathcal{F}_t) - \mathcal{G}_i^S(\mathcal{F}_s)\|. \tag{9}$$

where $\phi_i = softmax(I)$. $I$ represents the importance score acquired during the sampling of the N feature samples, as outlined in [32].

 

**Table 1: Comparison between our method and previous knowledge distillation methods on the KITTI dataset with PointPillars. The teacher and the student have 4.8M and 1.3M parameters, respectively. mAP indicates the mean average precision of moderate difficulty. The best and the sub-optimal results are marked in bold and blue, respectively.**

| Task | Method | Car | | | Pedestrians | | | Cyclists | | | mAP |
|------|--------|-----|---|---|-------------|---|---|----------|---|---|-----|
| | | Easy | Moderate | Hard | Easy | Moderate | Hard | Easy | Moderate | Hard | |
| BEV | Teacher w/o KD | 94.3 | 88.1 | 83.6 | 57.9 | 51.8 | 47.6 | 86.5 | 65.0 | 61.1 | 68.3 |
| | Student w/o KD | 92.4 | 88.2 | 83.6 | 53.0 | 47.9 | 44.1 | 81.8 | 63.1 | 59.0 | 66.4 |
| | *CRD*[22] | 92.7 | 87.8 | 83.2 | 56.6 | 50.4 | 46.8 | 80.3 | 61.9 | 57.9 | 66.7 |
| | *SE-SSD*[36] | 92.7 | 87.9 | 83.2 | 57.7 | 51.0 | 46.8 | 78.1 | 61.8 | 57.9 | 66.9 |
| | *Fitnets* [19] | 91.5 | 85.6 | 83.1 | 57.5 | 51.0 | 46.3 | 82.8 | 65.1 | 61.1 | 67.2 |
| | *FBKD*[34] | 92.3 | 85.7 | 83.0 | **59.7** | 52.0 | 47.6 | 71.0 | 64.3 | 60.5 | 67.5 |
| | *PATA* [31] | 92.7 | 88.0 | 83.6 | 56.7 | 50.9 | 47.3 | 81.4 | 64.4 | 60.5 | 67.7 |
| | *RDD*[13] | 92.4 | 88.0 | 83.5 | 57.9 | 51.6 | 47.6 | 82.3 | 64.6 | 60.8 | 68.2 |
| | *PointDistiller*[32] | 92.3 | 88.0 | 83.6 | 57.1 | 50.8 | 46.1 | **84.8** | **66.7** | 62.4 | 68.5 |
| | **+Ours** | **93.0** | **88.4** | **83.8** | 59.2 | **53.0** | **47.8** | 84.6 | **67.7** | **63.4** | **69.6** |
| 3D | Teacher w/o KD | 87.3 | 75.9 | 71.1 | 52.0 | 45.9 | 41.4 | 78.6 | 59.2 | 55.8 | 60.3 |
| | Student w/o KD | 87.4 | 75.9 | 71.0 | 48.2 | 43.0 | 38.7 | 74.1 | 57.2 | 53.3 | 58.7 |
| | *CRD*[22] | 85.6 | 74.2 | 71.0 | 49.5 | 43.5 | 39.0 | 76.4 | 58.4 | 54.7 | 58.7 |
| | *SE-SSD*[36] | 87.3 | 75.5 | 71.5 | 52.6 | 45.6 | 40.8 | 74.9 | 58.6 | 54.9 | 59.9 |
| | *Fitnets* [19] | 84.9 | 73.4 | 70.6 | 50.9 | 44.2 | 39.3 | 75.9 | 58.5 | 54.6 | 58.7 |
| | *FBKD*[34] | 87.5 | 75.8 | 71.6 | 53.4 | 45.8 | 40.9 | 76.1 | 59.0 | 55.2 | 60.2 |
| | *PATA* [31] | 87.6 | 75.7 | 71.4 | 51.0 | 44.8 | 40.7 | 74.4 | 57.8 | 54.2 | 59.5 |
| | *RDD*[13] | 87.5 | 76.0 | 71.4 | 50.7 | 44.0 | 40.0 | 80.0 | 60.1 | 56.2 | 60.9 |
| | *PointDistiller*[32] | 87.6 | 76.5 | 73.5 | 52.7 | 45.7 | 40.6 | 79.4 | 61.6 | 57.5 | 61.2 |
| | **+Ours** | **87.9** | **76.8** | **73.9** | **53.5** | **46.6** | **43.3** | **82.3** | **62.6** | **59.2** | **62.2** |

## 3.4 Loss Function

As shown in Figure 3, our model is optimized by task loss and KD loss. The overall objective function is calculated as follows:

$$\mathcal{L} = \mathcal{L}_D + \mathcal{L}_A + \mathcal{L}_{local} + \mathcal{L}_{task}. \tag{10}$$

The task loss will be different for different models. For voxels-based and point-based methods, task loss refers to [12, 30] and [21], respectively.

## 4 EXPERIMENTS

### 4.1 Dataset and metrics

Our experiments are conducted both on KITTI [8] and unScenes [2], which consist of samples that have both lidar point clouds and images. Our models are trained with only the lidar point clouds. The KITTI dataset consists of 7481 training samples and 7518 testing samples, with annotated objects in the car, pedestrians, and cyclists categories. The training samples were divided into 3712 training samples and 3769 testing samples. For KITTI, we report the average precision calculated by 40 sampling recall positions for BEV (Bird's Eye View) object detection and 3D object detection on the validation split. Following the typical protocol, the IoU threshold is set as 0.7 for class Car and 0.5 for class Pedestrians and Cyclists. Besides, the nuScenes dataset is another large-scale dataset used for autonomous driving, containing 1,000 driving sequences, where 700, 150, and 150 sequences are used for training,

validation, and testing, respectively. Each sequence is captured in approximately 20 seconds with 20 FPS using the 32-lane lidar. Its evaluation metrics are the average precision (mAP) and nuScenes detection score (NDS). NDS is a weighted average of mAP and true positive metrics which measures the quality of the detections in terms of box location, size, orientation, attributes, and velocity.

### 4.2 Implementation Details

We have evaluated our method in both voxels-based object detectors PointPillars [12] and CenterPoint [30], and the raw points-based object detector PointRCNN [21]. Following the PointPillars as the teacher network on KITTI, we use an AdamW optimizer [14] with a weight decay of 0.01 and a cyclic momentum update strategy to adjust the learning rate. We set 0.0001 for the initial learning rate, 1 for cyclic update time, and 0.90 for momentum. Following the Point-Pillars as the teacher network on unScenes, we use a step strategy to adjust the learning rate. The networks have been trained on RTX 3090 GPUs. All the experiments are conducted with mmdetection3d [5] and PyTorch[16]. We keep the evaluation settings in mmdetection3d as default. The teacher model is the origin model before compression. The student model shares the same architecture and depth as its teacher but with fewer channels. We have mainly compared our methods with previous knowledge distillation methods, including methods proposed by Fitnets[19], PATA[31], CRD[22],

**Table 2: Experimental results of our method for BEV (Bird-Eye-View) and 3D object detection on KITTI dataset, respectively. F indicates the number of float operations(/G) . P indicates the parameters (/M) of the detector. KD indicates whether our method is utilized. mAP indicates the mean average precision of moderate difficulty. The reported result in the first line of each detector is the performance of the teacher detector**

| Task | Model | F(/G) | P(/M) | KD | Car | | | Pedestrians | | | Cyclists | | | mAP |
|---|---|---|---|---|---|---|---|---|---|---|---|---|---|---|
| | | | | | Easy | Moderate | Hard | Easy | Moderate | Hard | Easy | Moderate | Hard | |
| BEV | PointPillars | 31.5 | 4.8 | × | 94.3 | 88.1 | 83.6 | 57.9 | 51.8 | 47.6 | 86.5 | 65.0 | 61.1 | 68.3 |
| | | 8.4 | 1.3 | × | 92.4 | 88.2 | 83.6 | 53.0 | 47.9 | 44.1 | 81.8 | 63.1 | 59.0 | 66.4 |
| | | | | ✓ | **93.0** | **88.4** | **83.8** | **59.2** | **53.0** | **47.8** | **84.6** | **67.7** | **63.4** | **69.6**$_{+3.2}$ |
| | | 2.3 | 0.3 | × | 91.3 | 84.8 | 82.2 | **50.1** | 44.4 | **41.6** | 74.2 | 56.1 | 52.5 | 61.8 |
| | | | | ✓ | **92.3** | **85.6** | **83.0** | 49.8 | **44.5** | 40.8 | **77.1** | **60.0** | **56.0** | **63.7**$_{+1.9}$ |
| | PointRCNN | 103.6 | 4.1 | × | 95.0 | 86.7 | 84.3 | 69.8 | 64.5 | 58.1 | 92.8 | 74.6 | 70.4 | 75.3 |
| | | 13.1 | 0.5 | × | 93.5 | 85.9 | 83.5 | 71.6 | 65.4 | 59.1 | 91.1 | 71.0 | 67.2 | 74.1 |
| | | | | ✓ | **94.3** | **86.7** | **84.1** | **75.2** | **68.2** | **62.3** | **93.9** | **71.8** | **68.1** | **75.8**$_{+1.7}$ |
| | | 6.8 | 0.3 | × | **95.8** | 85.4 | 81.7 | 72.9 | 65.5 | 58.6 | 91.8 | 69.3 | 65.9 | 73.4 |
| | | | | ✓ | 95.6 | **86.9** | **82.8** | **74.2** | **66.2** | **59.7** | **93.3** | **70.0** | **66.5** | **74.6**$_{+1.2}$ |
| 3D | PointPillars | 31.5 | 4.8 | × | 87.3 | 75.9 | 71.1 | 52.0 | 45.9 | 41.4 | 78.6 | 59.2 | 55.8 | 60.3 |
| | | 8.4 | 1.3 | × | 87.4 | 75.9 | 71.0 | 48.2 | 43.0 | 38.7 | 74.1 | 57.2 | 53.3 | 58.7 |
| | | | | ✓ | **87.9** | **76.8** | **73.9** | **53.5** | **46.6** | **43.3** | **82.3** | **62.6** | **59.2** | **62.2**$_{+3.5}$ |
| | | 2.3 | 0.3 | × | 83.1 | 69.8 | 65.4 | 44.0 | 38.7 | **35.3** | 70.9 | 52.1 | 48.7 | 53.5 |
| | | | | ✓ | **84.2** | **70.8** | **67.8** | **44.2** | **39.0** | 35.0 | **71.4** | **54.1** | **50.6** | **55.1**$_{+1.6}$ |
| | PointRCNN | 103.6 | 4.1 | × | 92.1 | 80.1 | 77.4 | 66.8 | 60.3 | 54.3 | 92.1 | 72.3 | 67.8 | 70.9 |
| | | 13.1 | 0.5 | × | 89.8 | 76.8 | 72.7 | 67.9 | 60.9 | 54.0 | 88.1 | **68.0** | **64.4** | 68.6 |
| | | | | ✓ | **91.5** | **77.2** | **73.2** | **70.0** | **63.1** | **57.1** | **91.0** | 67.6 | 62.8 | **69.3**$_{+0.7}$ |
| | | 6.8 | 0.3 | × | 89.8 | 75.3 | 70.7 | 68.7 | 60.7 | 53.4 | 91.1 | 67.2 | 63.9 | 67.7 |
| | | | | ✓ | **90.0** | **75.6** | **71.2** | **70.0** | **62.5** | **54.8** | **91.2** | **69.0** | **65.3** | **68.9**$_{+0.6}$ |

**Table 3: Ablation study on n KITTI dataset with 4× compressed PointPillars students. $\mathcal{L}_{global}$ and $\mathcal{L}_{local}$ indicate global distillation loss, including Distance-wise RKD loss and Angle-wise RKD loss, and local RKD loss, respectively. $\mathcal{L}_{knn}$ indicates we reproduced the loss of the method in the [32]. mAP indicates the mean average precision of moderate difficulty.**

| Model | Task | $\mathcal{L}_{global}$ | $\mathcal{L}_{local}$ | $\mathcal{L}_{knn}$ | Car | | | Pedestrians | | | Cyclists | | | mAP |
|---|---|---|---|---|---|---|---|---|---|---|---|---|---|---|
| | | | | | Easy | Moderate | Hard | Easy | Moderate | Hard | Easy | Moderate | Hard | |
| PointPillars | BEV | × | × | × | 92.4 | 88.2 | 83.6 | 53.0 | 47.9 | 44.1 | 81.8 | 63.1 | 59.0 | 66.4 |
| | | × | × | ✓ | 91.9 | 87.3 | 83.2 | 57.9 | 50.9 | 46.8 | 82.4 | 65.3 | 61.5 | 67.8$_{+1.4}$ |
| | | × | ✓ | × | 92.4 | 88.0 | 83.5 | 57.9 | 51.6 | 47.6 | 82.3 | 64.6 | 60.8 | 68.2$_{+1.8}$ |
| | | ✓ | × | × | 92.3 | 88.0 | 83.6 | 57.1 | 50.8 | 46.1 | **84.8** | 66.7 | 62.4 | 68.7$_{+2.3}$ |
| | | ✓ | ✓ | × | **93.0** | **88.4** | **83.8** | **59.2** | **53.0** | **47.8** | 84.6 | **67.7** | **63.4** | **69.6**$_{+3.2}$ |
| | 3D | × | × | × | 87.4 | 75.9 | 71.0 | 48.2 | 43.0 | 38.7 | 74.1 | 57.2 | 53.3 | 58.7 |
| | | × | × | ✓ | 85.2 | 73.9 | 70.7 | 52.6 | 45.5 | 40.8 | 76.4 | 57.4 | 53.9 | 59.7$_{+1.0}$ |
| | | × | ✓ | × | 85.2 | 75.2 | 68.7 | **53.7** | **47.0** | 42.4 | 75.3 | 60.8 | 56.9 | 61.0$_{+2.3}$ |
| | | ✓ | × | × | 84.9 | 75.9 | 68.9 | 51.4 | 45.4 | 41.4 | 78.5 | 61.3 | 57.8 | 60.9$_{+2.2}$ |
| | | ✓ | ✓ | × | **87.9** | **76.8** | **73.9** | 53.5 | 46.6 | **43.3** | **82.3** | **62.6** | **59.2** | **62.2**$_{+3.5}$ |

SE-SSD[36], FBKD[34], RDD[13], PointDistiller[32] on KITTI, and CRD[22], OFD[9], MTS[26], PointDistiller[32] on nuSecens.

## 4.3 Comparison with State-of-the-arts

**Results on KITTI.** Experiments of 4× compressed PointPillars on KITTI. Table 1 shows the performance between our method and previous knowledge distillation method for BEV detection and 3D detection, respectively. Our proposed H2RKD method shows good performance, outperforming most existing methods. It is observed that on BEV and 3D detection, our method outperforms the second-best knowledge distillation method by 1.1% and 1.0% moderate mAP, respectively. Furthermore, our method empowers the student detector to surpass the teacher detector, resulting in performance improvements of 1.3% and 1.9% in BEV and 3D detection,

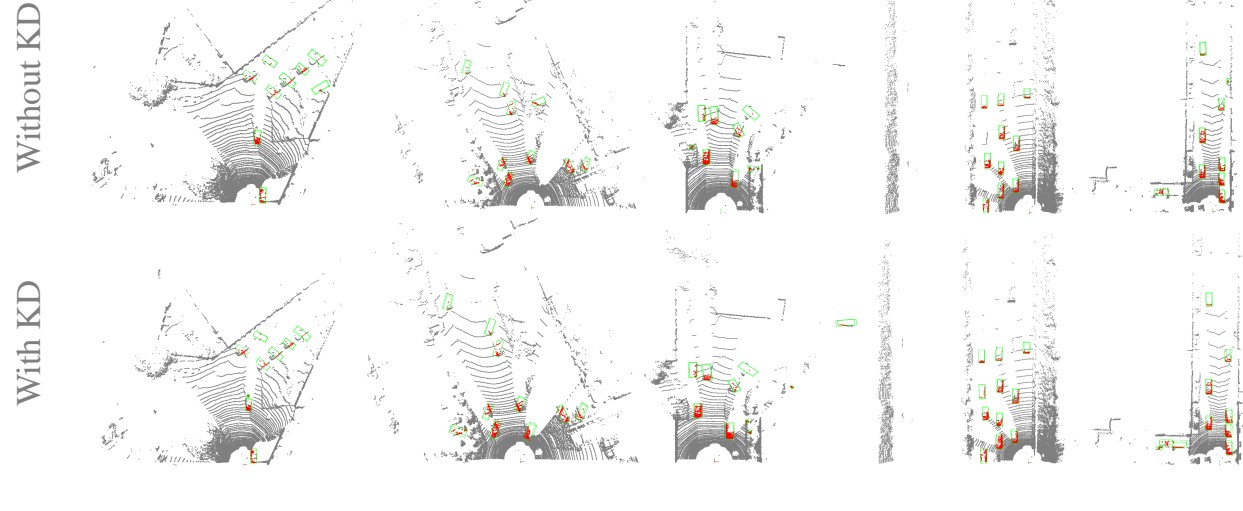

**Figure 4: Qualitative comparison between the detection results of students trained with and without knowledge distillation.**

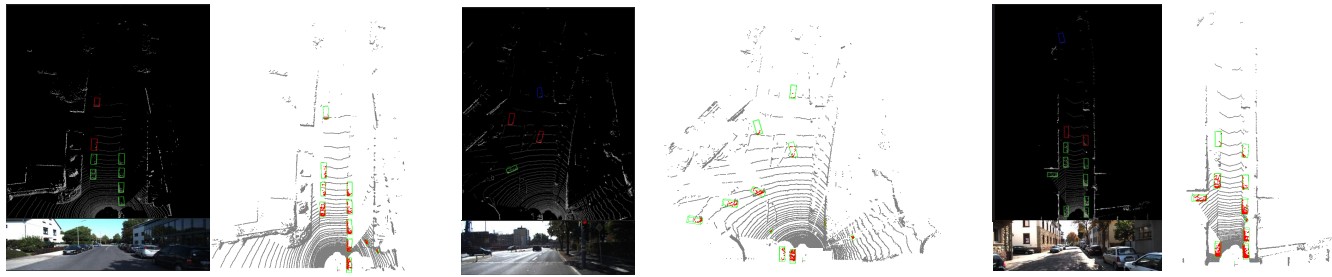

**Figure 5: On the BEV detection, the left side shows the ground truth and the original image, while the right side displays the detection results.**

respectively. Besides, our method attains the highest performance across all difficulty levels for car categories. It is worth mentioning that we performed optimally in each category on 3D detection. The above results validate the efficacy of our method in 3D Lidar-based object detection.

**Resluts on nuScenes.** Experiments of around 2× and 4× compressed PointPillars and CenterPoint on nuScenes are shown in Table 4. It is observed that our method leads to 0.5% and 0.6 % improvements on mAP and NDS on average, respectively for the compressed PointPillars model. Besides, the compressed Center-Point also shows 0.6 % improvements on NDS. These observations indicate that our method is also effective on the large-scale dataset.

In summary, the effectiveness of our method can be demonstrated on both two datasets.

## 4.4 Ablation Study

**Effects on different Detectors.** Table 2 shows the performance of the voxel-based and point-based detectors trained with and without our method for BEV detection and 3D detection, respectively. On average, 2.6% and 1.5% moderate mAP improvements can be

observed for the voxel and raw points-based detectors for BEV detection, respectively. Additionally, the voxel and raw points-based detectors exhibit moderate mAP improvements for 3D detection, averaging 2.6% and 0.7%, respectively. It demonstrates the advancements achieved by our approach in the voxel-based detector.

Specifically, in BEV detection, our method demonstrated superior performance with the 4× compressed and accelerated PointPillars detector student surpassing its teacher by 1.3% mAP. Additionally, the 8× compressed and accelerated PointRCNN detector student, trained with our method, outperformed its teacher by 0.5% mAP. Similarly, for the 3D detection of PointPillars detectors, the 4× compressed and accelerated student, trained with our method, achieved a notable improvement of 1.9% mAP compared to its teacher. Furthermore, the compressed and accelerated PointRCNN detector student, trained with our method, exhibited enhancements of 0.7% and 0.6% in mAP on 3D detection, respectively.

Consistent average precision boosts can be observed in the detection results of all categories. For instance, on BEV detection of 4× compressed PointPillars students, 0.3%, 4.4% and 3.7% mAP improvements can be observed for cars, pedestrians, and cyclists, respectively. Additionally, Consistent average precision boosts can

**Table 4: Experimental results on nuScenes dataset with Point-Pillars and CenterPoint. mAP indicates the mean average precision of moderate difficulty. NDS indicates nuScenes detection score. The best and the sub-optimal results are marked in bold and blue, respectively.**

| Model | F(/G) | P(/M) | Method | mAP(↑) | NDS(↑) |
|---|---|---|---|---|---|
| | 31.5 | 4.8 | Teacher w/o KD | 39.3 | 53.2 |
| | | | Student w/o KD | 36.0 | 50.5 |
| | | | CRD[22] | 35.7 | 50.4 |
| | 16.8 | 2.4(2x) | OFD[9] | 36.2 | 50.6 |
| | | | MTS[26] | 36.2 | 50.7 |
| | | | *PointDistiller*[32] | 36.5 | 51.0 |
| PointPillars | | | **Ours** | **37.2** | **51.6** |
| | | | Student w/o KD | 32.3 | 47.3 |
| | | | CRD[22] | 32.3 | 47.2 |
| | 8.4 | 1.3(4x) | OFD[9] | 32.4 | 47.6 |
| | | | MTS[26] | 32.5 | 47.8 |
| | | | *PointDistiller*[32] | 32.5 | 48.0 |
| | | | **Ours** | **32.7** | **48.6** |
| | 110.2 | 9.2 | Teacher w/o KD | 57.3 | 65.6 |
| | | | Student w/o KD | 55.2 | 64.0 |
| | | | CRD[22] | 55.6 | 64.4 |
| CenterPoint | 45.8 | 4.6(2x) | OFD[9] | 55.7 | 64.4 |
| | | | MTS[26] | 55.8 | 64.6 |
| | | | *PointDistiller*[32] | 56.2 | 65.1 |
| | | | **Ours** | **56.3** | **65.7** |

be observed in the detection results of all difficulties. For instance, on 3D detection of 16× compressed PointRCNN students, 0.5%, 1.3%, and 1.1% mAP improvements can be observed for easy, moderate, and hard difficulties, respectively.

In summary, these observations demonstrate that our method can successfully transfer teacher knowledge to the voxel-based and point-based student detectors. Furthermore, it also validates that our method is capable of learning an effective and lightweight 3D detector.

**Effects of CGD and SLD Modules.** Our H2RKD is mainly composed of two components, including collaborative global distillation (CGD) and separate local distillation (SLD). Ablation studies with 4× compressed PointPillars students on KITTI are shown in Table 3. We also compare with the local distillation in [12].

It is observed that on BEV detection and 3D detection, 2.3% and 2.2% mAP improvements can be obtained by only using CGD to distill the global relation within the point cloud, respectively. Moreover, the category of cyclists on easy demonstrates its optimal performance solely with the application of the CGD module.

Additionally, 2.2% and 2.3% mAP boosts can be gained by using SLD on BEV detection and 3D detection, respectively. Moreover, the pedestrian category on easy and moderate demonstrate the optimal performance solely using the SLD module, respectively. Furthermore, Compared to the approach of solely constructing the KNN graph, the SLD module enhances its performance by 0.4% and 1.3% in BEV and 3D detection, respectively. From the aforementioned

analysis, we observe that the SLD module plays a more significant role in 3D detection, indicating the importance of local structural relationships among point clouds.

In summary, these observations indicate that each module in H2RKD has its individual effectiveness and their merits are orthogonal. Besides, the CGD module proves to be more effective in BEV detection, whereas the SLD module demonstrates greater efficacy in 3D detection.

## 4.5 Visualization Analysis

In this subsection, we have visualized the detection results of the student model trained with and without our method, as shown in Figure 4. Furthermore, we visualized the detection results and conducted a comparison with the ground truth, as shown in Figure 5. Note that both student models are 4× compressed PointPillars trained on KITTI. The green boxes indicate the boxes of the model prediction. The visualization clearly demonstrates the strengths of our approach. Specifically, as shown in Figure 4, a model with our H2RKD exhibits the capability to detect object in distant regions thanks to our global relation distillation. when compared with the ground truth shown in Figure 5, our method proficiently identified the majority of objects and gained substantial local knowledge, as indicated by the red points.

## 5 CONCLUSION

In this paper, we aim to simultaneously transfer both homophily and heterophily relational knowledge of point clouds, enhancing intra-object similarity and inter-object discrimination. To this end, we propose a novel joint homophily and heterophily relational knowledge distillation method(H2RKD), which distills the relational knowledge by collaborative global distillation (CGD) and separate local distillation (SLD). Specifically, CGD transfers both distance-wise and angle-wise global relations, implicitly collaborating homophily and heterophily. To further transfer subtle correlations and differences, SLD explicitly distills local homophily and heterophily by reconstructed graphs, separately. Extensive experiments on KITTI and unScenes datasets demonstrate the effectiveness of the proposed H2RKD.

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
