# OpenReview forum: "Joint Homophily and Heterophily Relational Knowledge Distillation for Efficient and Compact 3D Object Detection"
_acmmm.org/ACMMM/2024/Conference — MM2024 Poster_

### Official Review · Reviewer_M7Bw · 2024-04-28

**Rating:** 4
**Confidence:** 4

**Summary:**

This paper proposes a new method called Joint Homophilic and Heterophilic Relational Knowledge Distillation (H2RKD) for distilling robust relational knowledge in point clouds to enhance similarities within objects and refine distinctions between objects. This unified strategy consists of integrating Collaborative Global Distillation (CGD), which is used to distill global relational knowledge across distance and angle dimensions, and Separate Local Distillation (SLD), which is used to centrally distill local relational dynamics. By seamlessly utilizing relational dynamics in point clouds, H2RKD facilitates comprehensive knowledge transfer and greatly improves 3D object detection.

**Strengths:**

The novelty of this paper is outstanding, making enough innovations to the technique of knowledge distillation in point clouds, by dividing and synthesizing the Homophily and Heterophily.

The paper is theoretically and experimentally detailed, through theoretical derivation of collaborative global distillation and separate local distillation, and embedded in a network of teachers and students.

The quality of the paper's presentation is excellent overall, but some of the expressions could be improved. In addition, there are several obvious factual errors in the paper.

The method proposed in this paper is an incremental component that needs to be embedded on existing methods to show specific gains, which may be a hindrance for real-world applications.

**Limitations:**

1. The font size in the illustrative figures in the paper needs to be adjusted for better presentation quality, especially in Figures 1 and 2.

2. The contribution mentions intra-object similarity and inter-object discrimination, which are widely used in self-supervised learning and transfer learning, however I did not find any substantial innovations made by the authors on them in the paper.

3. The framework in Figure 3 is in my opinion brilliant, however the captions could be improved. Suggest "The framework of the proposed joint homophily and heterophily relational knowledge distillation method (H2RKD) for 3D object detection." can be changed to "The framework of the proposed H2RKD for 3D object detection.". In addition, the icon in front of "Angle Relation" needs to be further adjusted.

4. There are irregularities in the case of symbols in Eqs. 7-9 in the paper, which need to be carefully reviewed by the authors.

5. Why are some of the methods in the results table set in italics?

6. There are some obvious factual or grammatical errors in the paper, such as:
"unScenes", "K-Nearest Neighbor (KNN)", "(1) Given or (2) Given", "Car or Pedestrians", "operations(/G) .", "y.L", etc. Note that some of the errors occur multiple times, the authors are asked to search globally in order to handle them appropriately.

7. Knowledge distillation and transfer learning mentioned in the paper are popular techniques, and the authors are suggested to supplement the methods and experiments in the related literature: "Self-supervised intra-modal and cross-modal contrastive learning for point cloud understanding" , TMM 2023; "M3SOT: Multi-Frame, Multi-Field, Multi-Space 3D Single Object Tracking", AAAI 2024; "Joint Semantic Segmentation using representations of LiDAR point clouds and camera images", INFFUS 2024.

**Suitability:**

2

---

### Official Review · Reviewer_g1wJ · 2024-05-23

**Rating:** 4
**Confidence:** 3

**Summary:**

This paper propose a novel Joint Homophily and Heterophily Relational Knowledge Distillation method for 3D Object Detection, which contains the Collaborative Global Distillation and Separate Local Distillation module to distill global relational knowledge and s subtle local correlations, respectively. Extensive experiments demonstrate the effectiveness of the proposed methods.

**Strengths:**

- The proposed method achieve SOTA performance against previous knowledge distillation methods on 3D detection.
- The motivation is clear, allowing the student to simultaneously learn from teacher about similar relationships (homogeneity) of the same object and different relationships (heterogeneity) of different objects.
- The experiments are solid.
- The paper is well-written and easy to follow.

**Limitations:**

- In the Collaborative Global Distillation, the proposed method uses distance-wise RKD loss and angle-wise RKD loss to distill the global relationship between students and teachers, which is very similar to the RKD method, and the authors need to give a detailed explanation about the difference with the previous method.
- The main motivation of the proposed method is to consider homogeneity and heterogeneity in relationship distillation. However, the motivation of the proposed Collaborative Global Distillation mainly focus on the global information transfer, and does not appear to consider homogeneity and heterogeneity, which may deviate from the original motivation. This point may weaken the contribution of this work.

**Suitability:**

2

---

### Official Review · Reviewer_9tjf · 2024-05-24

**Rating:** 4
**Confidence:** 3

**Summary:**

This paper proposes a novel methodology termed Joint Homophily and Heterophily Relational Knowledge Distillation (H2RKD) to distill robust relational knowledge in point clouds, thereby enhancing intra-object similarity and refining inter-object distinction. H2RKD consists of Collaborative Global Distillation (CGD) and Separate Local Distillation (SLD). The former is designed for distilling global relational knowledge across both distance and angular dimensions, and the latter is responsible for a focused distillation of local relational dynamics.

**Strengths:**

1. The authors focus on multi-dimensional relational distillation for 3D object detection, such as global and local, homophily and heterophily.
2. Experimental comparison is sufficient. The proposed method achieves SOTA performance.

**Limitations:**

1. There is confusion about inter-object and intra-object are used in the abstract, and inter-class and intra-class are used in the introduction.
2. A suggestion is that a subsection is needed to be added to organize the symbols.
3. Minor issue: L = I (i.) - A  in line 407.

**Suitability:**

3

---

### Official Review · Reviewer_33ZW · 2024-05-25

**Rating:** 4
**Confidence:** 1

**Summary:**

This paper focus on Knowledge Distillation for Lidar-based 3D Object Detection. The authors obverse that existing work is ineffective at distilling relationships with low similarity between features from the teacher to the student. Based on this finding, they proposed Collaborative Global Distillation and Separate Local Distillation to more comprehensively transfer the teacher's knowledge to the student. The experimental results on various datasets demonstrate the effectiveness of their methods.

**Strengths:**

In fact, I am not familiar with this field. Therefore, please take my opinion into consideration cautiously.

1. The writing is clear and easy to understand, the experiments are relatively thorough, and the results are quite satisfactory.
2. The motivation is reasonable, and the proposed methods are relevant to the motivation.
3.  The proposed methods seem to have a certain degree of novelty and are relatively comprehensive. For example, the authors approach relational distillation from both global and local perspectives. For the global aspect, they consider distances and angles between features for distillation, while for the local aspect, they take into account Homophily and Heterophily Relational Knowledge.

**Limitations:**

Since I am not familiar with this field, I cannot summarize any meaningful limitations. Please refer to the opinions of other reviewers.

**Suitability:**

2

---

### Meta-Review · Area_Chair_JvfG · 2024-07-03

**Recommendation:** Accept (Poster)
**Confidence:** 4

**Metareview:**

The paper has been thoroughly reviewed by four reviewers, all of whom agree on its acceptance. The strengths of this work lie in its novel insights into multi-dimensional relational distillation from both global and local perspectives, as well as homogeneity and heterogeneity. Furthermore, the method has been extensively validated, and its high performance demonstrates the effectiveness of the approach.

However, the authors must address several points, primarily related to writing and illustrations, before submitting the final version. Despite these shortcomings, the overall merits of the paper justify its acceptance, as recommended by the ACs. To ensure the quality of the work, the authors are advised to make significant improvements addressing all reviewers' concerns.